# Exploring the Potential of Artificial Intelligence in Pediatric Echocardiography—Preliminary Results from the First Pediatric Study Using AI Software Developed for Adults

**DOI:** 10.3390/jcm12093209

**Published:** 2023-04-29

**Authors:** Corina Maria Vasile, Xavier Paul Bouteiller, Martina Avesani, Camille Velly, Camille Chan, Zakaria Jalal, Jean-Benoit Thambo, Xavier Iriart

**Affiliations:** 1Department of Pediatric and Adult Congenital Cardiology, University Hospital of Bordeaux, 33600 Bordeaux, France; 2IHU Liryc—Electrophysiology and Heart Modelling Institute, Bordeaux University Foundation, 33600 Pessac, France; 3Department of Cardiology, Rythmology, CHU of Bordeaux, 33600 Pessac, France; 4Department of Cardiac, Thoracic, Vascular and Public Health Sciences, University of Padua, 235122 Padova, Italy

**Keywords:** echocardiography, pediatric cardiology, artificial intelligence, automated measurements, soft

## Abstract

(1) Background: Transthoracic echocardiography is the first-line non-invasive investigation for assessing pediatric patients’ cardiac anatomy, physiology, and hemodynamics, based on its accessibility and portability, but complete anatomic and hemodynamic assessment is time-consuming. (2) Aim: This study aimed to determine whether an automated software developed for adults could be effectively used for the analysis of pediatric echocardiography studies without prior training. (3) Materials and Methods: The study was conducted at the University Hospital of Bordeaux between August and September 2022 and included 45 patients with normal or near normal heart architecture who underwent a 2D TTE. We performed Spearman correlation and Bland-Altman analysis. (4) Results: The mean age of our patients at the time of evaluation was 8.2 years ± 5.7, and the main reason for referral to our service was the presence of a heart murmur. Bland-Altman analysis showed good agreement between AI and the senior physician for two parameters (aortic annulus and E wave) regardless of the age of the children included in the study. A good agreement between AI and physicians was also achieved for two other features (STJ and EF) but only for patients older than 9 years. For other features, either a good agreement was found between physicians but not with the AI, or a poor agreement was established. In the first case, maybe proper training of the AI could improve the measurement, but in the latter case, for now, it seems unrealistic to expect to reach a satisfactory accuracy. (5) Conclusion: Based on this preliminary study on a small cohort group of pediatric patients, the AI soft originally developed for the adult population, had provided promising results in the evaluation of aortic annulus, STJ, and E wave.

## 1. Introduction

Transthoracic echocardiography is the first-line non-invasive investigation for assessing pediatric patients’ cardiac anatomy, physiology, and hemodynamics due to its accessibility and portability. Currently, it is the most widely used imaging modality by cardiologists worldwide [1,2].

Frequently, performing echocardiography on pediatric patients can be challenging for physicians and sonographers due to poor pediatric compliance, restlessness, and crying of babies. This can lead to a significantly prolonged time to perform echocardiography, despite using highly trained staff and carefully optimized imaging protocols.

Over the past decade, artificial intelligence (AI) has produced several game-changing developments in healthcare, with diverse applications in assistance and patient management. AI has already been successfully integrated into many aspects of cardiology, including clinical risk stratification, image interpretation, prognosis, and treatment [3,4]. One of the potential benefits of using integrated artificial intelligence software in routine echocardiography could be an optimization of the scanning time by getting automatic measurements of the main cardiovascular structures, leaving time for the physician or the sonographer to focus on more specific anatomic or hemodynamic issues.

This study aimed to explore whether artificial intelligence software designed for adults could be used to interpret pediatric echocardiography without prior training for 23 specific echocardiographic parameters.

## 2. Materials and Methods

### 2.1. Study Design

This prospective study was performed between August and September 2022 in the Pediatric and Congenital Cardiology Department of the University Hospital of Bordeaux under the approval of the Institutional Review Board and Ethics Committee of Publication (approval no. CER-BDX-2023-03/13.03.2023). The children’s guardians were informed fully regarding the study protocol and had no objection to using the echocardiography assessment data for research purposes.

A set of echocardiographic images obtained on 45 patients aged between 0 to 18 was analyzed by a junior physician, a senior physician, and the AI software provided by Ligence Heart Company. Twenty-three features were assessed through echocardiography for each patient. We compared the agreement between methods and observers. Our primary purpose was to establish whether the Ligence Heart software validated for analyzing adult echocardiography could be transposed to pediatric patients.

The ultrasound scans were performed manually by two medical practitioners, a senior with more than ten years of experience and a junior with less than two years of experience. The studies were also automatically analyzed by the Artificial Intelligence software provided by the Ligence Heart Company.

### 2.2. Study Population

The cohort group enrolled in the study was comprised of 45 children, aged between 0–18 years, who were referred to our clinic for transthoracic echocardiography assessment in the context of a simple screening examination for a systolic murmur, chest pain, or follow-up check after atrial septum defect (ASD) closure and patent ductus arteriosus (PDA) closure. All patients evaluated echocardiographically post-ASD and PDA closure have undergone a percutaneous interventional procedure for correction.

The inclusion criteria included children aged between 0–18 years. From our study, we excluded patients with complex congenital heart diseases, as we aimed to have a homogenous group of patients with typical or near-normal heart anatomy.

### 2.3. Echocardiographic Assessment

The images were acquired using Philips Epiq 7 (the S9-2 probe for newborns, toddlers, and older children X5-1 probe) and Siemens Acuson SC2000 (the 4V1C probe for older children and the 8V3 probe for newborns and toddlers) ultrasound systems. All patients were in sinus rhythm during image acquisition within a normal heart rate range. The images were transferred in DICOM format to the Ligence Heart server via a secure local area network. Each study contained multiple DICOM instances. All patient’s data were anonymized before transfer to the Ligence Heart software. Further analysis was performed using the Ligence Heart software.

2D TTE studies were acquired manually by the senior and the junior cardiologists using manual functionalities of the post-processing software (Ligence Heart version 3.5.0, Ligence, UAB, Vilnius, Lithuania).

Before each acquisition, image settings were optimized by modifying the gain, compress, and time gain compensation controls to achieve the highest possible frame rate.

We performed a complete examination and assessed the following parameters: in the parasternal long axis (PLAX), in apical 4 chambers (A4CH), and in apical 2 chambers (A2CH) according to international guidelines. The assessed parameters are presented in Table 1.

### 2.4. Automatic Measurements

This study utilized the Ligence Heart software (v. 3.5.0), developed by the Ligence company in Vilnius, Lithuania. The software is specifically designed for analyzing echocardiographic images of adult patients, enabling the detection, measurement, and calculation of various cardiac structure and function parameters. The software is built to imitate the typical steps taken by a cardiologist, including image view recognition, cardiac cycle phase detection, and measurement calculation. The software uses deep learning technology to achieve its clinical results, identifying the cardiac cycle by tracking the endocardial border rather than analyzing ECG waves. In this study, automated segmentations and measurements were performed on all 23 traits previously evaluated by two clinicians. The Ligence Heart software’s analysis included the automated identification of view modes, such as PLAX, A4CH, and A2CH, in the DICOM instances and the automated identification of end-systolic and end-diastolic frames within a DICOM instance. The software then performed automated segmentations and measurements on each DICOM instance, as listed in Table 1.

Figure 1 shows the automated evaluation of left ventricular measures for PLAX.

### 2.5. Statistical Analysis

Within each feature, extreme outliers from the software were removed before analysis. These outliers were selected based on a visual inspection of each histogram, and only extreme outliers were removed (i.e., values that can be clearly stated, without any doubt, as erroneous). We compared both AI–Senior measurements and Senior–Junior measurements. Patients were separated into two groups based on age: 0 to 9 and 9 to 18. Scatter plots were drawn for each feature, and the related Spearman correlation coefficient was calculated, although we acknowledge the limitation of such an indicator in the case of method comparisons [5,6]. Bias, Percentage of Error (PE), and tolerance intervals were assessed using Bland-Altman Analysis [7]. As Francq et al. [8] stated, a tolerance interval is “an interval where a given proportion of the population should lie, on average.” We calculated tolerance intervals [9,10] instead of the traditionally used limits of agreement because they are exact. They avoid estimating confidence intervals of the upper and lower bound which can be misleading [10].

As defined by Bland & Altman [7], bias was calculated as the mean difference between observations. The standard deviation (SD) of the differences was also computed.

The Tolerance Intervals (TI) were calculated as follows:$$TI=bias\pm t_{0.975,n-1}\times SD\sqrt{1+\frac{1}{n}}$$

The percentage of error was calculated as follows:$$PE\left(\%\right)=100\%\times t_{0.975,n-1}\times \frac{SD\sqrt{1+\frac{1}{n}}}{\mu }$$
where $t_{975,n-1}$ is the 97.5% percentile of the Student’s t distribution (with n − 1 degrees of freedom) and µ is the mean value of the observations. we slightly modified the original formula [11,12,13] to be consistent with our tolerance intervals. As recommended, a threshold of 30% in the error percentage was used to assess the agreement between measurements [11,12,13].

We used a linear regression model to investigate the relationship between the percentage of error of the software-senior physician comparison as the dependent variable and the age group (under or higher than 9) and the percentage of error of the comparison between senior and junior physicians as independent variables).

Finally, to investigate the software’s ability to perform measurements, the number of missing values related to the software measurements was compared between age groups using a χ^2^ test.

All analyses were conducted with python V3.8 and R V4.1.

## 3. Results

The baseline characteristics of the study patients are summarized in Table 2. The mean age of our patients at the time of evaluation was 8.2 ± 5.7 years, and most were males (60%). The main reason for referral to our service was the presence of a heart murmur (n = 13), followed by arrhythmia (n = 8).

Patients were classified into two subgroups: the first group (patients aged between 0–9 years, 60%) and the second group (patients over 9 years, 40%). The decision was made because pediatric anatomy differs from adults and to verify if the dedicated artificial intelligence software for adults has better reproducibility for pediatric patients older than 9 years.

We observed strong Spearman correlations (Figure 2) between the junior and senior pediatric cardiologists for most parameters ranging from 0.8 to 0.95. Only correlations of two values among 23 were below 0.70 (r = 0.53 for Fractional Area Change and r = 0.68 for Ejection Fraction).

A strong correlation was noted among cardiologists for LV measurements in the PLAX view and aortic annulus diameter, ranging from 0.84 to 0.96. Correlations between AI and the senior cardiologist for these parameters generally had a slightly lower tendency but remained higher than 0.57.

Although V max TR and STJ assessments showed a strong correlation between cardiologists, correspondences between AI and the senior cardiologist tended to have a weaker correlation (<0.5) for these parameters.

When we split the data into two age groups (under 9 years old and older than 9), the same pattern persisted with good correlations between physicians and much more variability between software and physicians according to the studied parameters. Although it should be noted that, even for adults, the software was not validated on all these studied parameters (Figure 3).

There were very strong correlations between cardiologists and AI software and cardiologists when assessing parameters in the A4CH view (LV, LA, RV, RA). We observed strong correlations between cardiologists and strong for AI and cardiologists for LV volumes in A2CH. The same tendency was for spectral Doppler parameters. The weakest correlations between AI and cardiologists were for FAC and LVEF. Interestingly that for the same variables, the weakest correlations were between cardiologists.

Bland Altman’s analysis showed that globally there is a good agreement between both physicians with a mean error of 35.2% (below 9 years old) and 35% (above 9 years old), and the error value is lower than the 30% cut-off in respectively 12 and 13 measurements among 23. However, it should be emphasized that even between physicians, the agreement could be weak; for example, the error is 90.0% (below 9) and 83.6% (above 9) for the ESV LV 4CH (Figure 4), and 4 (below 9) and 6 (above 9) features have a percentage of error higher than 50%.

When we compared the automatic software and the senior physician, the agreement was better for the subgroup of patients older than nine years old (mean percentage of error of 53.4% for patients under 9 years old and 45.8% for patients older than 9 years old). However, for two assessed echocardiographic parameters in the youngest group and four traits for the group >9 years old, the error was below 30% (Figure 5).

The scatter plot for the percentage of error of the senior-software comparison against the senior-junior comparison for each feature illustrates that the mean percentage error is slightly higher for the youngest group. Still, this difference is insignificant (Estimated parameters βabove9 = −3.95, F = 1.0, *p* = 0.32). Additionally, there is a considerable difference in the percentage of error between software and physician compared to both physicians; the percentage of error is significantly higher among software and physician (Estimated parameters βsoft-physician = 14.5, F = 13.5, *p* = 0.0004). There are 3 groups defined: in dark blue, there is a good agreement both between senior physician and the software and between physicians; in light blue, the agreement is good between physicians only, and in the white background, there is a weak agreement both between senior physician and the software and between physicians.

Our study reported good agreement for all patients’ aortic annulus and E wave measurements. When we divided the patients into two groups, the agreement was much better for the subgroup, which included patients over 9 years old, achieving an excellent result for the automatic evaluation of sinotubular junction (STJ) and EF (ejection fraction).

There is a significant difference (χ^2^ = 14.8, *p* = 0.0001) regarding the number of missing values in the software measurements between age groups (Table 3. The software returned significantly more null (aka missing) values for the youngest group (18.3% of the values vs. 9.4%)

The proportion of missing values returned by the software was not homogenous across the features, for example, “AOR ANNULUS” or “ESD” respectively, 47.8% and 36.9% of the returned values were missing, whereas for “FAC”, “E WAVE” and “EF” only 9.5%, 11.1%, and 9.1% were missing.

The average time for performing a full scan on a pediatric population was approximately 15 min for both junior and senior doctors.

## 4. Discussion

This is the first study to test automated software’s feasibility and variability on a cohort of pediatric patients. The present study supports evidence suggesting that automated software designed for adults could be easily transposed to assessing young patients with a reduced analysis time.

Artificial Intelligence is evolving to address more complex imaging needs, as demonstrated by improvements in image acquisition quality and the commercial availability of automated measurements.

Due to the increasing number of patients with cardiovascular disease and the growing demand for echocardiographic studies, there is a capacity bottleneck between patient volume and the number of experts available to interpret echocardiographic studies in the laboratory and at the point of care [14]. Therefore, AI, which can offer sustainable solutions to reduce workloads (e.g., aid in the repetitive and tedious tasks involved in echocardiography measurement and interpretation), should be viewed as an opportunity to deal with this expanding demand on the condition that AI can reach a similar or higher measurement’s accuracy than physicians [15].

These advances could help physicians to improve their productivity, work, and diagnostic performance. Artificial intelligence can reduce echocardiography’s cost while improving its reliability and quality [15]. The automatic software we used for our pediatric patients was already validated by Karužas et al. [16] for evaluating the aortic valve in adult patients. Assessment of left ventricular volume and function was one of the first applications of artificial intelligence to minimize error and reduce operator subjectivity [17,18,19,20]. Research methods have progressed, and recently Knackstedt et al., demonstrated that left ventricular ejection fraction and longitudinal strain could be analyzed in approximately 8 s using machine learning methods [21].

Transthoracic echocardiographic imaging is the most common non-invasive cardiac procedure performed. However, image quality varies substantially from patient to patient and is also operator dependent, which increases interobserver variability. We also observed a significant variation in image interpretation between physicians according to the measured feature, with a percentage of error ranging from approximately 10% (when assessing E wave and aortic annulus) to 90% (when assessing volumes from A2CH and A4CH). Inaccuracy in measurements and differences in echocardiography interpretation are frequent and often associated with conflicting interpretations in echocardiography reports [22]. In most cases, such errors can be related to physician fatigue, impaired attention, memory, and executive function that diminish the reader’s recall and alert to detail [15].

In the study performed by Bobbia et al. [23], they compared, for the first time, the interpretability of images acquired by highly and less-trained echocardiography emergency physicians using a pocket ultrasound device in a prehospital setting. Less experience significantly reduces the interpretability of focus echocardiography performed under these conditions. Therefore, the physician’s experience may sometimes affect the diagnostic performance.

Based on the reported data of Arbic N. et al. [24], the average time for performing echocardiography by a pediatric cardiologist is, on average, more than 20 min [24], based on the experience of the CHU Bordeaux center, while the average time for completing echocardiography by a sonographer is at least one hour, based on the data of the SickKids Hospital, Canada [24]. The average time for performing the measurements by our pediatric cardiologist was on average 15 min, while for the automatic soft was 55 ± 11 s. The measurements performed with artificial intelligence software may decrease the echocardiographic evaluation time since it does not require image freezing and manual delineations of anatomic structures using a built-in keyboard. Therefore, using AI for analyzing echocardiography could allow a significant decrease in the analysis time that could benefit physicians if the automatic measurement is at least as accurate as the human-based analysis.

Our study reported good agreement between physicians and AI software for aortic annulus measurements and E wave for all patients, without any differences depending on age. When we splitted the patients into two groups, we achieved good agreement for patients older than 9 years old when we assessed the sinotubular junction (STJ) and the ejection fraction (EF). In a recent study, Karužas et al. [16] used the same software on a cohort of 58 adults to evaluate aortic measurements. They obtained promising results, with high accuracy, using the AI software for the evaluation of aortic measurements in 2D TTE PLAX view: the difference in variabilities between human operators and AI-based software measurements were insignificant, the correlation of AI-based software with the expert cardiologist was higher than between junior cardiologist and expert cardiologist. In their study, no difference in terms of accuracy between the AI and the physician was found for the measurement of the aortic annulus, aortic sinus, proximal ascending aorta, and sinotubular junction measurements. Interestingly, these features were not used for training the AI, thus, it demonstrates the scalability of the software. Together with the result of Karužas et al. [16], our findings show that the AI could allow physicians to perform the automatic measurements of the aortic annulus, E wave, STJ, and EF in a pediatric population. However, further investigation is needed to understand why the AI software performs poorly in the younger patient for these features, this might be due to the inherent anatomical difference in heart structure between the younger versus the older patient, but it could also be an artifact of the software that can be reduced with proper training.

For most of the parameters assessed, an agreement was good between physicians (PE < 30%), but weaker between physicians and software (IV S, PW D, IVS D, PW S, V MAX TR) (see Figure 6). The lack of scalability of the AI may be due to the intrinsic characteristics of pediatric images for these features. Physicians might overcome this difficulty, but the AI software will need extensive research and training in order to reach good accuracy for these parameters. Based on this hypothesis, it would be possible to train artificial intelligence to achieve good accuracy for these features: either a software could be trained from scratch, but would require a full development process, or an already trained software, such as Ligence Heart, could be optimized by transfer learning. It would require a big database of pediatric echocardiography images in both cases. However, given that agreement between clinicians is good, an increase in performance could be expected.

A poor agreement was found in all comparisons for the last set of features (FAC, RA AREA, ESA RV, EDV LV 4 CH, ESV 2 CH). Actually, no agreement was found even between cardiologists when evaluating volumes and areas. Therefore, the pattern of these parameters is very tenuous. Training the software for these features would require a large dataset with proper labeling thus, for now, it seems quite unrealistic. For these specific types of measurements, AI software cannot yet be used, and it requires further studies on a larger group of patients of different ages.

To sum up, we found that AI software previously trained on adult echocardiography could be successfully transferred for analyzing pediatric images. However, not all features are suitable for AI assessment without additional training. Nevertheless, for at least four features (E wave, aortic annulus, EF, and STJ) AI software could be directly applied to the measurements. AI might reach a better accuracy for the other parameters after proper training.

## 5. Limitations

In this study, we also encountered some limitations. The major was that we used special adult software on a pediatric population, including newborns, with different vessel and heart chamber sizes. Another potential impediment could be the small number of patients included.

In addition, all our evaluation data came from a single center in France, and echocardiographic images were obtained using ultrasound machines from two manufacturers (Siemens and Philips). However, it should be noted that this was the first study to test the feasibility of this new automated software and assess its ability to compete with conventional measurements calculated by human experts.

Another limitation was that the automated software could not return values for all parameters assessed whenever the echocardiographic image did not have the best quality.

Some differences may interfere because cardiologists chose ES/ED by visual eye or ECG, while AI chose ES/ED frames independently based on endocardial segmentation.

## 6. Conclusions

Artificial intelligence automatic softs could help clinicians to save time for repetitive, low-level, routine tasks such as measurements, standardization of data preparation, and quality control, thus allowing more time for higher levels of interpretation, patient care, and medical decision-making. Artificial intelligence software also helps improve the accuracy and consistency of interpretations.

Based on the findings of this preliminary study on a limited number of pediatric patients, the automatic soft could help pediatric cardiologists assess the aortic annulus and E wave for patients of all ages, and for patients older than 9 years old, it could be useful even for evaluating STJ and EF in normal or near-normal heart structure.

## 7. Future Perspectives

Non-invasive assessment of the heart by echocardiography provides us with representations of the interplay between cardiac anatomy and physiology.

Often, we perform a reductionist approach to identify critical features of an ultrasound that are most associated with the underlying pathophysiology of the heart relative to clinical signs and symptoms [25]. While easy to perform, this approach must consider the complexity and nuances of echocardiography that could improve the detection of physiological and anatomical changes in cardiac health and disease and their association with clinical outcomes.

In recent years, machine learning has successfully identified more subtle and complex patterns in echocardiographic data [26]. This ability to learn and classify patterns, coupled with modern computing power, allows us to improve our understanding of cardiac anatomy and physiology and optimize and automate logistical tasks in the clinical echocardiography laboratory.

## Figures and Tables

**Figure 1 jcm-12-03209-f001:**
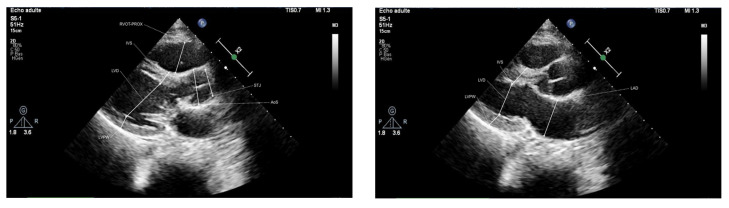
Fully automated measurements made by the Ligence Heart system. From left to right: PLAX in ED, PLAX in ES. (*PLAX—parasternal long axis, ED—end-diastole, ES—end-systole*).

**Figure 2 jcm-12-03209-f002:**
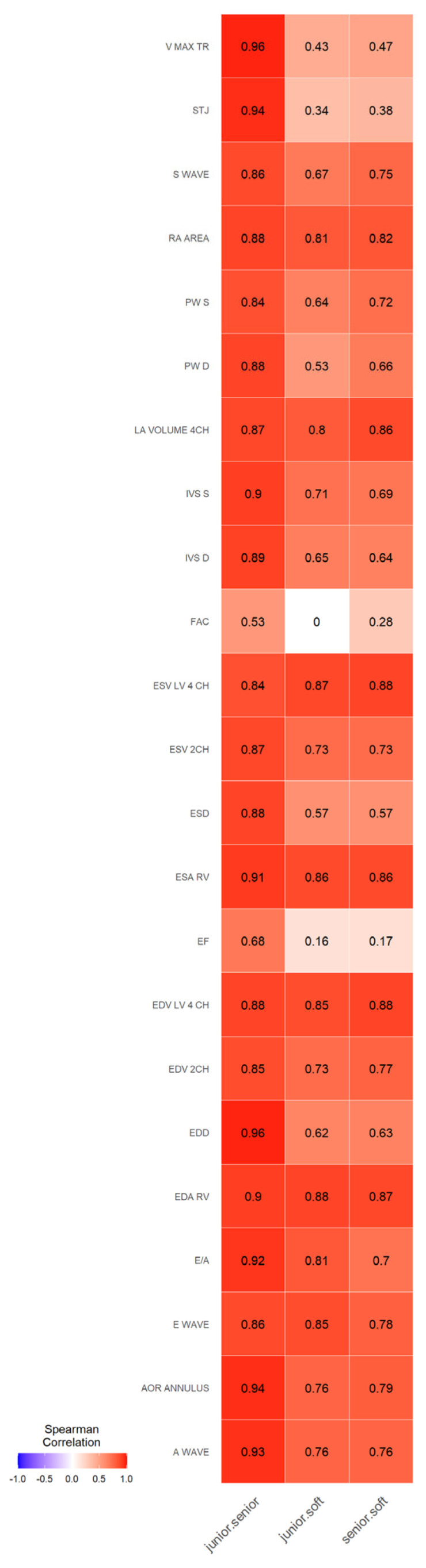
Spearman correlation between each measurement of junior and senior physicians, junior physician and software senior and software. Color is related to the value of the correlation coefficient.

**Figure 3 jcm-12-03209-f003:**
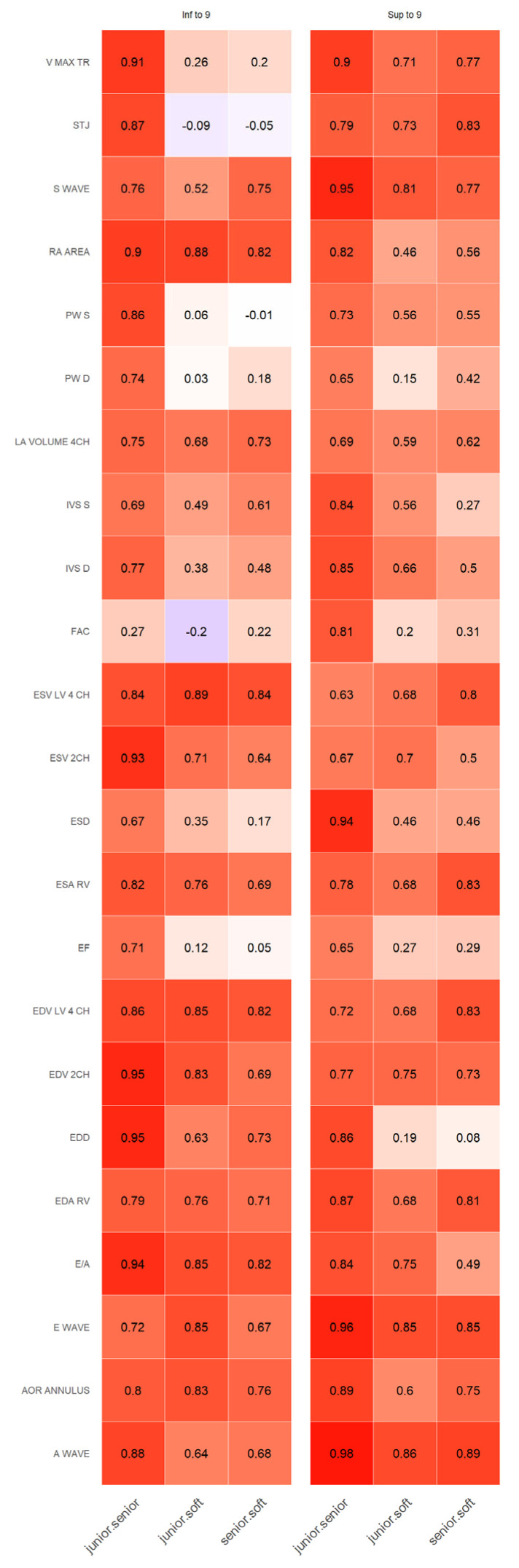
Spearman correlation between each measurement for the two subgroups. Color is related to the value of the correlation coefficient.

**Figure 4 jcm-12-03209-f004:**
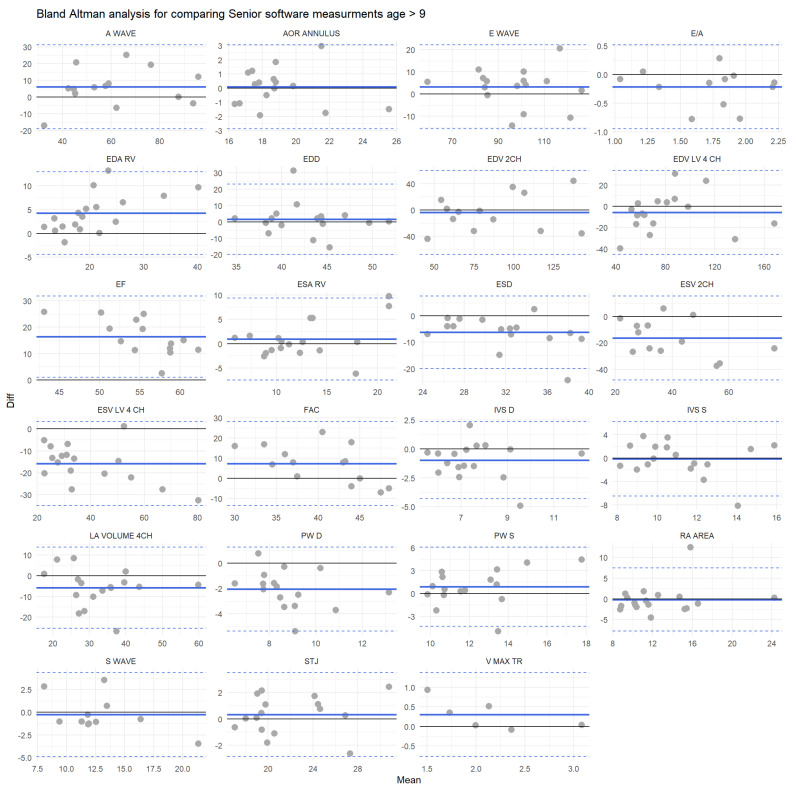
Bland Altman plots for each feature for individuals older than 9. Measurements’ differences between the senior physician and the software are plotted against the mean value of the two measurements. Black solid lines represent the equation y = 0; solid blue lines illustrate the mean bias and the dashed blue lines are the boundaries of the tolerance intervals.

**Figure 5 jcm-12-03209-f005:**
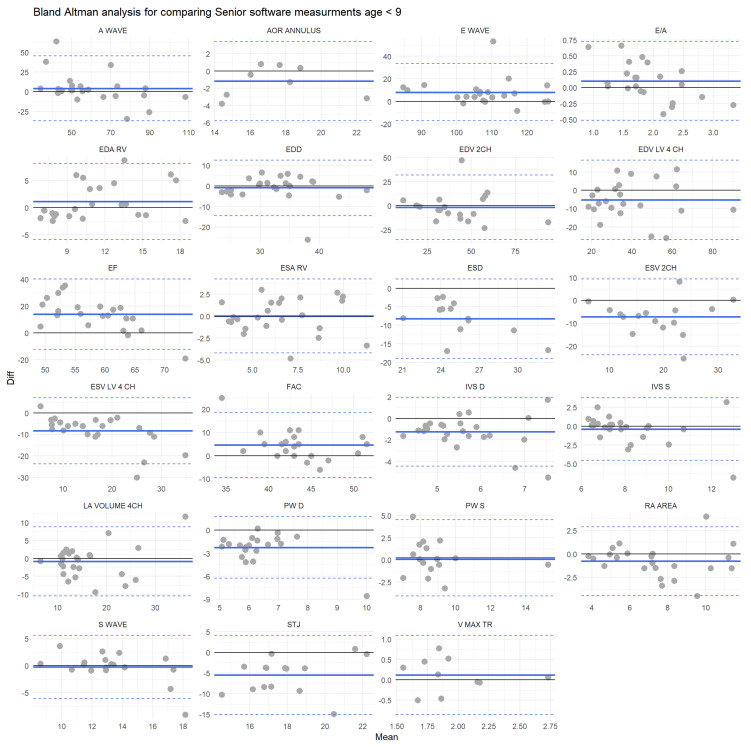
Bland Altman plots for each feature for individuals younger than 9. Measurements’ differences between the senior physician and the software are plotted against the mean value of the two measurements. Black solid lines represent the equation y = 0, solid blue lines illustrate the mean bias, and the dashed blue lines represent the tolerance intervals’ boundaries.

**Figure 6 jcm-12-03209-f006:**
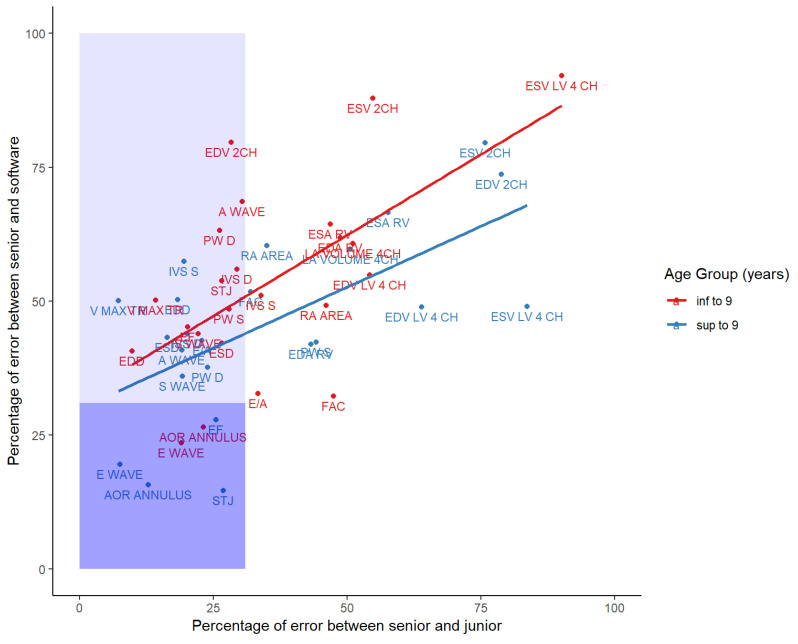
Scatter plot of the Percentage of Error (PE) for the senior—software comparison against the senior-junior comparison for each feature. In red is the PE for the patients under 9 and in blue is the Pe for the patients above 9. The straight line shows the regression against the two values. Shape in light blue corresponds to the area where the PE for the senior-junior comparison is lower than the 30% cut-off. Shape in dark blue corresponds to the area where the PE for the junior-senior comparison and the PE for the senior—software comparison are lower than the 30% cut-off.

**Table 1 jcm-12-03209-t001:** Echocardiographic sections and assessed parameters manually and automatically.

View	Measurement	Abbreviation
**Parasternal long axis (PLAX)**	LV posterior wall thickness in systole	PW S
	LV posterior wall thickness in diastole	PW D
	LV interventricular septum dimensions in systole	IV S
	LV interventricular septum dimensions in diastole	IV D
	LV end-diastolic diameter	LVEDD
	LV end-systolic diameter	LVESD
	aortic annulus	AOR ANNULUS
	sinotubular junction	STJ
**Apical 4 chambers (A4CH)**	Transmitral E velocity	E wave
	Transmitral A velocity	A wave
	E/A ratio	E/A
	Wave S right ventricle	Wave S
	Tricuspid regurgitation maximum velocity	V MAX TR
	Fractional area change of RV	FAC
	LV End-Systolic Volume	ESV LV 4 CH
	LV End-Diastolic Volume	EDV LV 4 CH
	Left Ventricular Ejection Fraction	EF
	Left Atrial Volume	LA VOLUME
	Right Atrial Area	RA AREA
	Right Ventricle End-Diastolic Area	EDA RV
	Right Ventricle End-Systolic Area	ESA RV
**Apical 2 chamber (A2CH)**	LV End-Systolic Volume	ESV 2 CH
	LV End -Diastolic Volume	EDV 2 CH

**Table 2 jcm-12-03209-t002:** Baseline characteristics (N = 45).

Characteristic	Values
**Quantitative variable: mean ±SD, [min–max]**	
Age (years)	8.2 ± 5.7, [0.2–18.0]
Height (cm)	123 ± 36, [60–182]
Weight (kg)	31 ± 21, [5–85]
BSA (m^2^)	1.0 ± 0.48, [0.28–2.01]
**Qualitative variable: no. (%)**	
Gender (Female)	20 (44%)
Age group (<9 years old)	27 (60%)
Murmur	13 (28.8%)
Post ASD closure	7 (15.5%)
Chest pain	6 (13.3%)
Arrhythmia	8 * (17.7%)
PDA	4 (8.88%)
Post PDA closure	3 (6.66%)
Restrictive VSD	4 (8.88%)

Note: Unless otherwise noted, quantitative variables are presented as mean ± standard deviation, and qualitative variables are presented as numbers of patients with percentages in parenthesis. BSA = body surface area; ASD = atrial septal defect; PDA = patent ductus arteriosus; VSD = ventricular septal defect. * bradycardia (n = 3) and paroxysmal supraventricular tachycardia (n = 5).

**Table 3 jcm-12-03209-t003:** Contingency table for the missing values in the software measurements between the age group.

Age Group	Number of Missing Values	Number of Calculated Values
<9 years old	109	486
≥9 years old	38	368

## Data Availability

The data presented in this study are available on request from the n corresponding author.

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
