# Peer review of "Exploring the Potential of Artificial Intelligence in Pediatric Echocardiography—Preliminary Results from the First Pediatric Study Using AI Software Developed for Adults"

_jcm, 2023, doi:10.3390/jcm12093209_

Round 1

Reviewer 1 Report

It is a very interesting manuscript with a novel  and very interesting topic.

As already by the authors mentioned the study includes hust a small cohort. AI learning propably would perform better for a bigger patient cohort, as already mentioned in the paper. 

In addition, no agreement was found even between cardiologists when evaluating volumes and areas. The problem with volumes and areas are the formula that are being used for calculation of these. The formulas vor areas and volumes includes mathematical geometric asumptions like the Simpson and/or Teichholz formula, which are as a matter of fact derived by a healthy grown up population, where a small measurement error will lead to bigger error due to the formula used, as they include squaring. Thus it would be interesting to calculate true volumes by 3D imaging if avaible. 

Author Response

Dear Reviewer,

We greatly value your appreciation for our work and for taking the time to revise it.

Q1. As already by the authors mentioned the study includes hust a small cohort. AI learning propably would perform better for a bigger patient cohort, as already mentioned in the paper. 

A1. As we mention it in the manuscript, these are the preliminary results of this study. We used the software only on a small cohort to observe if the Ligence Heart soft, developed initially for adults, could also be used for children. We are considering more patients in the study to see if we can obtain better results for a larger cohort group.

Q2: In addition, no agreement was found even between cardiologists when evaluating volumes and areas. The problem with volumes and areas are the formula that are being used for calculation of these. The formulas vor areas and volumes includes mathematical geometric asumptions like the Simpson and/or Teichholz formula, which are as a matter of fact derived by a healthy grown up population, where a small measurement error will lead to bigger error due to the formula used, as they include squaring. Thus it would be interesting to calculate true volumes by 3D imaging if avaible. 

A2: Unfortunately, for the small cohort, we could not calculate the volumes by 3D, but we are planning to include this measurement for the ongoing larger cohort study.

Kind regards,

Corina Vasile

Reviewer 2 Report

The paper entitled “Exploring the potential of artificial intelligence in pediatric echocardiography. Preliminary results from the first pediatric study using AI software developed for adults” by Vasile and colleagues is an original article about a study whose aim was to investigate the use of Artificial Intelligence (AI) software for the interpretation of pediatric echocardiography and to establish the feasibility, utility, and variability of automated software using AI designed for adults in a cohort group of 45 pediatric patients. 

The paper is interesting, novel and well written. However, the authors might consider the following comments:

·      In the abstract of the paper, the aim of the study is not clear.

·      The authors state that performing a pediatric echocardiography with complete anatomic and hemodynamic assessment is time-consuming. When patients are infant or children, time-consuming is the acquisition of the images and not the measurement of different structures of the heart or of functional parameters. This is due to due the poor compliance, restlessness and crying of babies.

·       The authors conclude that the automatic soft could help pediatric cardiologists assess the aortic annulus and E wave for patients of all ages, and for patients older than 9 years old, it could be useful even for evaluating sinotubular junction (STJ) and ejection fraction (EF) in normal or near-normal heart structure. The possibility of entrusting the echocardiographic diagnosis to AI alone seems rather remote.

·      Unfortunately, it seems that at the present the AI is not able to make a precise diagnosis or reliable in measuring the echocardiographic parameters required by a routine examination.

·       The authors state that the main reason for referral to their service was the presence of a heart murmur(n=13), followed by arrhythmia (n=8). Which kind of arrhythmia?  

·      About patients enrolled in the study after atrial septal defect or ductus arteriosus closure, did they undergo cardiac surgery or a percutaneous procedure?

·      In the Table 1, in the column Measurement, it is better to replace “LV posterior wall diameter in systole” and “LV posterior wall diameter in diastole” with “LV posterior wall thickness in systole” and “LV posterior wall thickness in diastole”.

Author Response

Dear Reviewer,

We greatly value your appreciation for our work and for taking the time to revise it.

Q1: In the abstract of the paper, the aim of the study is not clear.

A1: We revised our abstract presentation according to your suggestion as follows:

“ Aim: This study aimed to determine whether an automated software developed for adults could be effectively used for the analysis of pediatric echocardiography studies without prior training.”

Q2: ·      The authors state that performing a pediatric echocardiography with complete anatomic and hemodynamic assessment is time-consuming. When patients are infant or children, time-consuming is the acquisition of the images and not the measurement of different structures of the heart or of functional parameters. This is due to due the poor compliance, restlessness and crying of babies

A2: We revised our statement. Regarding the reduced time for the assessment, while using the AI software, the physician does not have to freeze the image and perform the measurements; it’s enough only to take more loops and the soft perform independently.

Introduction: “ Frequently, performing echocardiography on pediatric patients can be challenging for physicians and sonographers due to poor pediatric compliance, restlessness, and crying of babies.  This can lead to a significantly prolonged time to perform echocardiography, despite using highly trained staff and carefully optimized imaging protocols.”

Discussion : "The measurements performed with artificial intelligence software may decrease the echocardiographic evaluation time since it does not require image freezing and manual delineations of anatomic structures using a built-in keyboard."”

Q3: ·       The authors conclude that the automatic soft could help pediatric cardiologists assess the aortic annulus and E wave for patients of all ages, and for patients older than 9 years old, it could be useful even for evaluating sinotubular junction (STJ) and ejection fraction (EF) in normal or near-normal heart structure. The possibility of entrusting the echocardiographic diagnosis to AI alone seems rather remote.

A3: We consider artificial intelligence approaches have been developed to be integrated into medical practice to facilitate some tasks, not to replace the doctor. Also, with sufficient data training, the software may be able to perform more measurements accurately.

Q4: Unfortunately, it seems that the AI is currently unable to make a precise diagnosis or reliable in measuring the echocardiographic parameters required by a routine examination.

A4: The poor accuracy for some parameters is probably because the present software has been trained and developed for the echocardiographic assessment of adults. This was the first study using this software on pediatric patients without prior training.  The cardiac anatomic features’ size difference between adults and children, particularly in the younger group, may explain the lack of accuracy compared to expert pediatric cardiologists.

Q5: The authors state that the main reason for referral to their service was the presence of a heart murmur(n=13), followed by arrhythmia (n=8). Which kind of arrhythmia?  

A5: We added in our baseline table the missing data. Of the 8 patients with arrhythmia, 3 presented for bradycardia and 5 for paroxysmal supraventricular tachycardia. All patients were in sinus rhythm during the image acquisition in a normal range of heart rhythm.

Q6: About patients enrolled in the study after atrial septal defect or ductus arteriosus closure, did they undergo cardiac surgery or a percutaneous procedure?

A6: All the patients that presented for follow-up after ASD and PDA closure had undergone the percutaneous procedure.

We added in Section 2.2. “All patients evaluated echocardiographically post-ASD and PDA closure have undergone a percutaneous interventional procedure for correction.”

Q7: In the Table 1, in the column Measurement, it is better to replace “LV posterior wall diameter in systole” and “LV posterior wall diameter in diastole” with “LV posterior wall thickness in systole” and “LV posterior wall thickness in diastole”.

A7: Thank you for the suggestion. We revised accordingly.

Kind regards,

Corina Vasile

Round 2

Reviewer 2 Report

Thank you for giving me the opportunity to review the revised version of the paper.

I would like to thank the authors for considering my suggestions.

Author Response

We thank the reviewer once again for the valuable remarks.